# Stoichiometric characteristics of woody plant leaves and responses to climate and soil factors in China

**Xiangguang Duan** [ID]*

School of Tourism, Xinyang Normal University, Henan, China

* m18238802921@163.com

**Data Availability Statement:** All relevant data are included in the paper and its supporting information file

**Funding:** The authors did not receive specific funding for this work

## Abstract

The main research content of the field of ecological stoichiometry is the energy of various chemical elements and the interaction between organisms and the environment throughout ecological processes. Nitrogen and phosphorus are the main elements required for the growth and development of plants and these also form the constituent basis of biological organisms. Both elements interact and jointly regulate the growth and development of plants, and their element ratios are an indication of the nutrient utilization rate and nutrient limitation status of plants. Previous research developed a general biogeography model of the stoichiometric relationship between nitrogen and phosphorus in plant leaves on a global scale. Further, it was shown that the relative rate of nitrogen uptake by leaves was lower than that of phosphorus, and the scaling exponent of nitrogen and phosphorus was 2/3. However, it is not clear how the stoichiometric values of nitrogen and phosphorus, especially their scaling exponents, change in the leaves of Chinese woody plants in response to changing environmental conditions. Therefore, data sets of leaf nitrogen and phosphorus concentrations, and nitrogen to phosphorus ratios in Chinese woody plants were compiled and classified according to different life forms. The overall average concentrations of nitrogen and phosphorus in leaves were $20.77 \pm 8.12$ mg g$^{-1}$ and $1.58 \pm 1.00$ mg g$^{-1}$, respectively. The contents of nitrogen and phosphorus in leaves of deciduous plants were significantly higher than those of evergreen plants. In leaves, life form is the main driving factor of nitrogen content, and mean annual temperature is the main driving factor of phosphorus content; soil available nitrogen is the main driving factor of the nitrogen to phosphorus ratio. These values can be used for comparison with other studies. In addition, the scale index was found to be significantly different among different life forms. The scaling exponents of N-P of woody plants of different life forms, such as trees, shrubs, evergreen, deciduous, and coniferous plants are 0.67, 0.72, 0.63, 0.72, and 0.66, respectively. The N-P scaling exponent of shrubs was higher than that of trees, and that of deciduous plants was higher than that of evergreen plants. These results suggest that the internal attributes of different life forms, the growth rate related to phosphorus, and the relative nutrient availability of soil are the reasons for the unsteady relationship between nitrogen and phosphorus in leaves.

**Competing interests:** The authors have stated that there is no competing interest

## 1 Introduction

The discipline of ecological stoichiometry combines the basic principles of ecology and chemometrics to study the energy balance and the balance of multiple chemical elements in the ecosystem; it is an important method to study the distribution of plant nutrient elements [1–3]. Ecological stoichiometry mainly emphasizes the relationship between the three main constituent elements of living organisms—carbon, nitrogen and phosphorus—it is also a theory for analyzing the impact of the mass balance of multiple chemical elements on ecological interactions [4]. It organically integrates research at different levels of ecology (i.e., individuals, populations, communities, ecosystems, and global levels) [5]. As the basic constituent elements of living organisms, nitrogen (N) and phosphorus (P) are considered important limiting factors in terrestrial ecosystems; they also play an important role in plant growth and physiological metabolic processes [6, 7]. Identifying the N and P nutrient status in plants is crucial for understanding their growth status and responding to changes in the conditions of the surrounding environment [8, 9]. In recent years, considerable research has been conducted on the stoichiometric characteristics of plant leaves, and the relationships between leaf stoichiometric characteristics and biological and abiotic factors have been explored at the regional [10, 11], national [12, 13], and global scales [14, 15]. Despite extensive research on the relationship between traits in the plant kingdom, our understanding of the coordination of traits at the level of woody organisms lags behind that of herbaceous species [16]. Previous studies have identified a set of key functional traits that summarize the form and functional spectrum of the entire plant kingdom, with leaf economy and plant size being the main trait axes supporting life cycle strategies [16]. Due to the size, life span, individual development and unique structural characteristics of woody plants, compared with herbaceous plants, trees have obvious characteristics and face new abiotic stress [17]. Our current understanding of dominant trait patterns in plants fundamentally overlooks the significant energy investment structure unique to large woody species [17]. Many studies have validated some assumptions about the relationship between nitrogen, phosphorus, and environmental factors at regional and global scales [10]. However, at present, there is still a lack of unified conclusions about the stoichiometric distribution and spatial variation pattern of plant leaves [18]. For example, the element content of plant leaves does not significantly change with increasing altitude[18–21]. Previous research also showed that the N: P ratio in plant leaves does not significantly change with increasing latitude [18, 22, 23]. In addition, data collected from published literature showed that with increasing mean annual temperature (MAT) and mean annual precipitation (MAP), the N and P concentrations in leaves decrease toward the equator, while the N: P ratio increases toward the equator [22, 23]. N concentration and N: P ratio are significantly higher in senescent leaves, and the P concentration decreases with increasing MAT and MAP [24]. It has also been shown that there is no relationship between the N: P ratio and average annual precipitation [25]. Based on these findings, scholars have put forward many theoretical hypotheses to explain the spatial variation law of the diversity of plant leaf chemometric characteristics; hypotheses such as the temperature biogeochemical hypothesis and the temperature plant physiological hypothesis were proposed [25].

In addition to the influence of climatic factors on the N and P stoichiometry of plants, it has also been shown that N and P stoichiometry of leaves is related to soil conditions [26–29]. The reason is that the aboveground and underground components of ecosystems are closely related [30, 31]. However, at present, the experimental evidence on controlling N and P stoichiometric change factors mainly focuses on climatic factors, while the evidence from soil available

nitrogen (AN) and soil available phosphorus (AP) is very limited [32, 33]. Research showed that when many plant life forms are included in a research study, life forms are often identified as the key driving factor for the variation of plant leaf chemometric characteristics, which weakens the influence of climatic factors and soil characteristics to a large extent [34, 35]. In addition, significant differences were found in the stoichiometric characteristics and spatial distribution patterns of leaves among plants of different life forms. For example, the N and P contents in the leaves of herbaceous plants are usually significantly higher than those of trees. The element content in the leaves of deciduous tree species is significantly higher than that of evergreen tree species, and a significant difference was found in the altitude distribution pattern of leaf stoichiometry between the two types of trees [23, 36]. Research has identified species variation as the main influencing factor of the N: P ratio of leaves in tropical areas [25]. Biological factors, such as leaf traits [10, 37], plant age [38], and functional groups [39] are all related to plant N and P chemometrics. Therefore, to understand the stoichiometric characteristics and control factors of plant N and P, comprehensive analysis of many aspects is needed.

In addition to N and P contents, as well as the N: P ratio, the metrological characteristics of leaf N and P contents can also be quantitatively represented as $N = \beta P^{\alpha}$. In this equation, $\alpha$ and $\beta$ represent the slope (scaling exponent) and "altitude" or intercept (normalization constant) of the logarithmic linear regression curve of the N concentration and P concentration of leaves, respectively [40, 41]. The scaling exponent essentially reflects the relative accumulation rate of N relative to P and is considered key for predicting plant and ecosystem functions [41–43]. The scaling exponent is also considered to be vital for predicting plant and ecosystem functions [4, 44, 45]. However, it has been shown that the size of the scale index is not consistent, for example, the proportion index of woody and herbaceous plants is on average two-thirds and three-quarters, respectively. Based on extensive collection of leaf N and P concentrations worldwide, it has also been suggested that the proportion index of N to P is about two-thirds [40, 41, 46]. In addition, several other studies have shown that the scaling exponent is about 0.67 [23], 0.72 [47], 0.78 [48], or 1.0 [49]. According to the most comprehensive data known so far (based on 9300 observations of leaf N and P concentrations), there are similar two-thirds scaling exponents in biological communities, taxonomic departments, and angiosperm life forms; therefore, a general 2/3 power law ($N = P^{2/3}$) was proposed for major plant groups and biological communities [41]. If a simple stoichiometric ratio controls the relationship between leaf stoichiometry and metabolism, despite differences in specific case studies, this would be very important. Moreover, the N-P power function relationship of leaves is based on multi-sample mixed statistical analyses at the population, community, or ecosystem levels. In essence, how sample size, sample source/composition, climatic factors, and soil factors affect the N-P power function relationship remains unknown.

To test the universality of leaf N and P chemometric relationships and the influencing factors of the N-P scaling exponent in Chinese woody plants, in this paper, the overall chemometrics of leaf N and P are discussed first, as they can be used as a basis for comparison with other studies [14]. Next, the N and P contents and scale indices of different life forms were determined, and it was examined whether numerical changes in stoichiometric characteristics are related to climatic conditions and soil factors [22, 50, 51]. In addition, the numerical change patterns of different site scaling exponents are also discussed to test whether the overall scale relationship between N and P masked significant differences in site correlation [14].

## 2 Materials and methods

To carry out the above research, a large, geographically comprehensive data set of paired distribution of N and P concentrations in Chinese woody plant leaves was collected. This data set

comprises regional and site level records, including as many relevant variables as possible. Only the records of N and P paired concentrations of leaves with detailed location information were used, while all records without location information or with unpaired concentrations were excluded. Via the authors' own on-site sampling, open TRY dataset (https://www.try-db.org) [52], and published literature, a total of 413 genera and 939 species were collected. Duplicate records were deleted. A total of 10,719 pairs of Chinese woody plant leaf N and P data were collected. The research area covers almost all provinces of China (Fig 1). There were 67 sites with more than 10 but less than 20 records, and 112 sites with more than 20 records. The Flora of China (http://frps.eflora.cn/) and Wikipedia (https://en.wikipedia.org/wiki) were used for identification and classification verification of plant functional groups. Broad-leaved forest data refers to data other than coniferous forests. LN (leaf N content) was measured using the methods described by He et al. (2006) [35]. LP (leaf P content) was measured after $H_2SO_4$-$H_2O_2$-HF digestion via the ammonium molybdate/stannous chloride method. Stoichiometric ratios (N:P) were calculated based on dry mass [53].

The statistical analysis was similar to that used in a previous study [41]. For data analysis, the N and P concentration data of all plants in the data set were log10 converted, and then, SMA regression was used to determine the N and P ratio at two levels (i.e., species functional groups and individual points). Therefore, plants were first divided into shrubs and trees, and then into needles, deciduous trees, and evergreen trees. Finally, the N-P scale relationship (N > 10 records) was analyzed, and the numerical changes of scaling exponents at different stations were quantified. A likelihood ratio test was used to evaluate the heterogeneity of the SMA regression index within the above analysis level [54]. In addition, a general linear regression was conducted using the statistical software package R 4.0.2 [55] to explore the change of the scale index with average N and P concentrations and N: P ratios.

## 3 Results

### 3.1 Stoichiometric characteristics of leaves with different life forms

Using 10,719 pairs of leaf N and P concentration data, the characteristics of large-scale leaf N and P stoichiometry were analyzed from the aspect of functional groups. The geometric mean values of leaf N and P concentrations, as well as N: P mass ratio of combined data were $20.77 \pm 8.12$ mg g$^{-1}$, $1.58 \pm 1.00$ mg g$^{-1}$, and $17.28 \pm 22.40$ respectively, but these values differed significantly among different functional groups (Table 1). The average LN (LP) contents of trees, shrubs, evergreen, deciduous, needle leaves, and broad-leaved were $20.56 \pm 8.12$ ($1.56 \pm 1.00$), $21.57 \pm 8.06$ ($1.60 \pm 1.00$), $17.43 \pm 6.77$ ($1.28 \pm 0.92$), $23.81 \pm 7.95$ ($1.80 \pm 1.00$), $14.98 \pm 5.66$ ($1.54 \pm 0.96$), $20.77 \pm 8.12$ ($1.58 \pm 1.00$), respectively. The N:P ratios of trees, shrubs, evergreen, deciduous needle leaves, and broad-leaved were $17.22 \pm 9.59$, $16.92 \pm 8.31$, $18.33 \pm 10.32$, $16.11 \pm 7.69$, $12.94 \pm 9.45$, and $17.28 \pm 22.41$, respectively. The LN of shrub plants were significantly higher than tree plants. The LN and LP of deciduous woody plants were significantly higher than evergreen woody plants. The LN of broad-leaved woody were significantly higher than coniferous woody plants. At different research sites, leaf N and P chemometrics were also statistically different. The geometric mean concentration and N: P mass ratio of leaf N and P positions were significantly different in 179 site records (more than 10 records). LN ranged from 2.45 to 36.06 mg g$^{-1}$, LP ranged from 0.17 to 4.61 mg g$^{-1}$, and N: P ranged from 4.10 to 40.27.

### 3.2 Leaf stoichiometry changes with environmental gradient

The LN of arbor, shrub, evergreen, deciduous, coniferous and broad-leaved woody plants was significantly correlated with MAT, and decreased with increasing MAT ($R$ = -0.40, $P$ < 0.001;

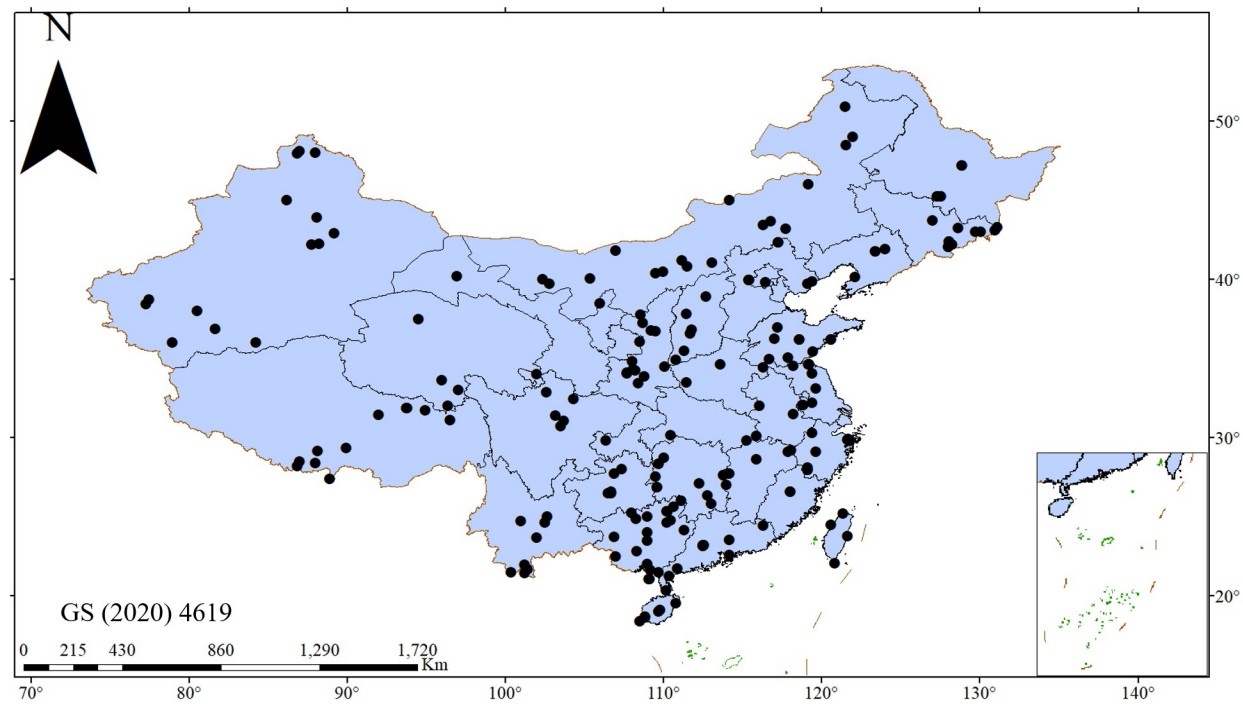

**Fig 1. Chinese sampling locations of leaf nitrogen and phosphorus levels for all species in this study.**

$R$ = -0.29, $P$ < 0.001; $R$ = -0.21, $P$ < 0.001; $R$ = -0.13, $P$ < 0.001; $R$ = -0.13, $P$ < 0.01; $R$ = -0.35, $P$ < 0.001). The LP of arbor, shrub, evergreen, deciduous, coniferous and broad-leaved woody plants was significantly correlated with MAT, and decreased with increasing MAT ($R$ = -0.34, $P$ < 0.001; $R$ = -0.29, $P$ < 0.001; $R$ = -0.12, $P$ < 0.001; $R$ = -0.25, $P$ < 0.001; $R$ = -0.16, $P$ < 0.001; $R$ = -0.31, $P$ < 0.001). The leaf N: P of arbor, shrub, evergreen, deciduous, coniferous and broad-leaved woody plants was significantly correlated with MAT, and increased significantly with increasing MAT ($R$ = 0.21, $P$ < 0.001; $R$ = 0.23, $P$ < 0.001; $R$ = 0.08, $P$ < 0.001; $R$ = 0.27, $P$ < 0.001; $R$ = 0.27, $P$ < 0.001; $R$ = 0.21, $P$ < 0.001) (Fig 2). However, the correlation between phosphorus content in leaves of evergreen woody plants and N: P and MAT was found to be weak.

The LN of arbor, shrub, evergreen, deciduous, coniferous and broad-leaved woody plants was significantly correlated with MAP, and decreased with increasing MAP ($R$ = -0.36, $P$ < 0.001; $R$ = -0.35, $P$ < 0.001; $R$ = -0.24, $P$ < 0.001; $R$ = -0.10, $P$ < 0.001; $R$ = -0.10,

**Table 1. Summary of the statistics of N and P concentrations and N: P ratios in leaves of different functional groups of terrestrial plants.** Multiple comparisons were made for each group of trees and shrubs, evergreen and deciduous, coniferous and broad-leaved. Mean represents the geometric mean, and n represents the number of observations. Different letters (such as a and b) represent significant differences ($p$ < 0.05).

| Form | n | LN | mean(mg g$^{-1}$) | LP | mean(mg g$^{-1}$) | N:P | |
|---|---|---|---|---|---|---|---|
| Woody plant | 10719 | 20.77 | ±8.12 | 1.58 | ±1.00 | 17.28 | ±22.40 |
| Tree | 5143 | 20.56 | ±8.12 b | 1.56 | ±1.00 a | 17.22 | ±9.59 a |
| Shrub | 4970 | 21.57 | ±8.06 a | 1.60 | ±1.00 a | 16.92 | ±8.31 a |
| Evergreen woody | 4344 | 17.43 | ±6.77 b | 1.28 | ±0.92 b | 18.33 | ±10.32 a |
| Deciduous woody | 5772 | 23.81 | ±7.95 a | 1.80 | ±1.00 a | 16.11 | ±7.69 b |
| Coniferous woody | 603 | 14.98 | ±5.66 b | 1.54 | ±0.96 a | 12.94 | ±9.45 b |
| Broad-leaved woody | 10116 | 20.78 | ±8.12 a | 1.58 | ±1.00 a | 17.28 | ±22.41 a |

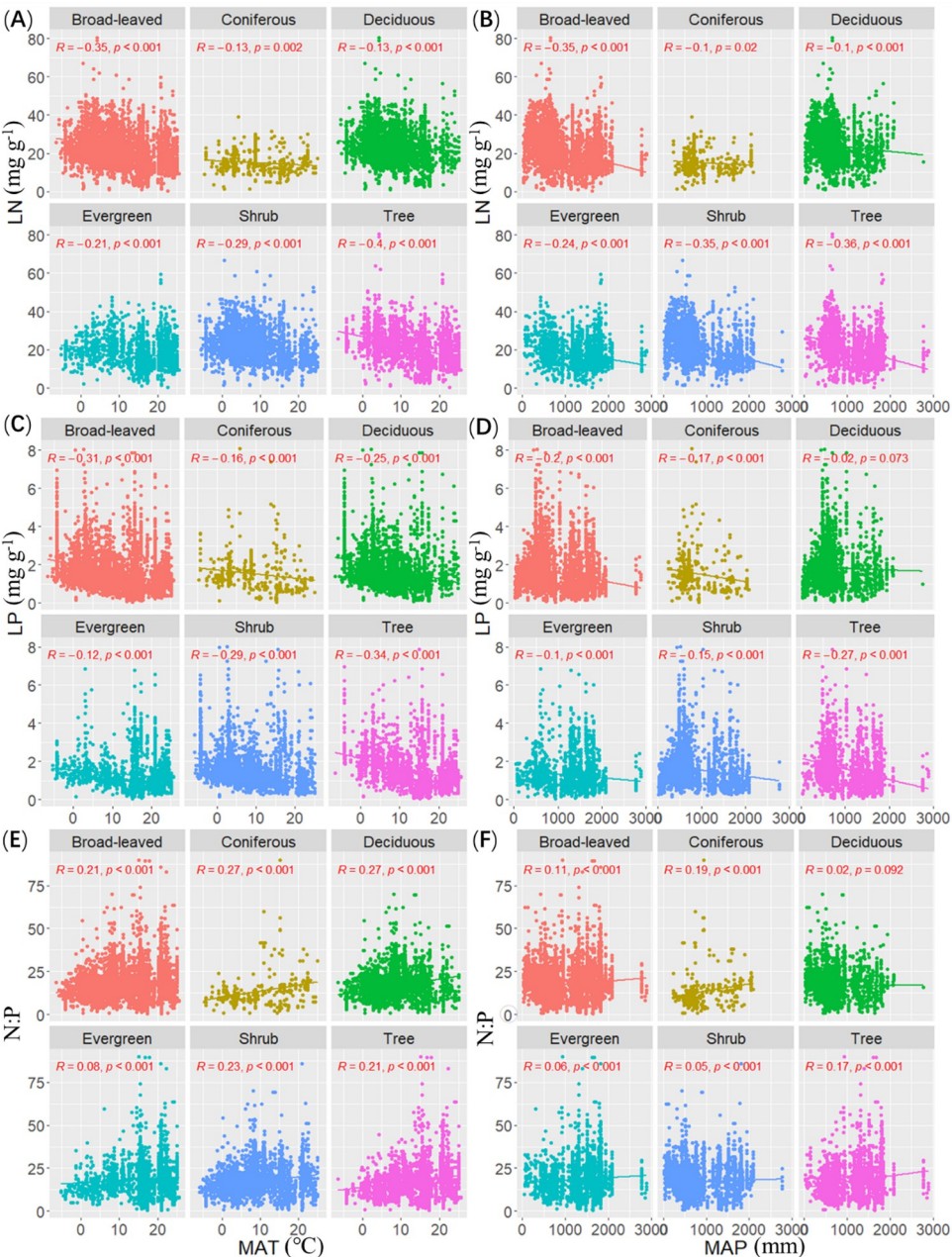

**Fig 2. Relationship between stoichiometry of leaves of different life forms and climatic factors.**

$P < 0.001$; $R = -0.35$, $P < 0.001$). The LP of arbor, shrub, evergreen, coniferous and broad-leaved woody plants was significantly correlated with MAP, and decreased with increasing MAP ($R = -0.27$, $P < 0.001$; $R = -0.15$, $P < 0.001$; $R = -0.10$, $P < 0.001$; $R = -0.17$, $P < 0.001$; $R = -0.20$, $P < 0.001$). However, MAP had no significant effect on LP in deciduous woody plants and had a weak correlation with evergreen woody leaves. The leaf N: P of arbor, shrub, evergreen, coniferous and broad-leaved woody plants was significantly correlated with MAP, and increased with increasing MAP ($R = 0.17$, $P < 0.001$; $R = 0.05$, $P = 0.027$; $R = 0.06$, $P = 0.003$; $R = 0.19$, $P < 0.001$; $R = 0.11$, $P < 0.001$) (Fig 2). However, the correlation between N: P and

MAP of evergreen woody plants and shrubs was found to be weak, and MAP had no significant effect on N: P of deciduous woody plants.

The LN of arbor, deciduous and broad-leaved woody plants was significantly correlated with AN, and increased with increasing AN ($R$ = 0.20, $P$ < 0.001; $R$ = 0.05, $P$ < 0.001; $R$ = 0.07, $P$ < 0.001). The LN of shrubs was significantly correlated with AN, but decreased with increasing AN ($R$ = -0.04, $P$ < 0.05). AN had no significant effect on evergreen and coniferous woody plants. The LP of arbors, shrubs, evergreen, deciduous and broad-leaved woody plants was significantly correlated with AN, and increased with increasing AN ($R$ = 0.20, $P$ < 0.001; $R$ = 0.18, $P$ < 0.001; $R$ = 0.06, $P$ < 0.001; $R$ = 0.21, $P$ < 0.001; $R$ = 0.19, $P$ < 0.001). However, no significant correlation was found between the LP of coniferous woody plants and AN, and a weak correlation was found between AN and evergreen plants. The leaf N:P of arbors, shrubs, evergreen, deciduous, coniferous and broad-leaved woody plants was significantly correlated with AN, and decreased with increasing AN ($R$ = -0.14, $P$ < 0.001; $R$ = -0.26, $P$ < 0.001; $R$ = -0.07, $P$ < 0.001; $R$ = -0.28, $P$ < 0.001; $R$ = -0.08, $P$ < 0.05; $R$ = -0.20, $P$ < 0.001). However, a weak correlation was found between AN and N: P of evergreen plants (Fig 3).

The LN of arbor, deciduous and broad-leaved woody plants was significantly correlated with AP, and increased with increasing AP ($R$ = 0.21, $P$ < 0.001; $R$ = 0.06, $P$ < 0.001; $R$ = 0.10, $P$ < 0.001). However, AP had no significant effect on the LN of shrubs, evergreen plant and coniferous woody plants, and had a weak correlation with the LN of deciduous woody plants (Fig 3).

The LP of trees, shrubs, evergreen, deciduous, and broad-leaved woody plants was significantly correlated with AP, and increased with increasing AP ($R$ = 0.18, $P$ < 0.001; $R$ = 0.07, $P$ < 0.001; $R$ = 0.08, $P$ < 0.001; $R$ = 0.10, $P$ < 0.001; $R$ = 0.13, $P$ < 0.001). The LP of trees, shrubs, evergreen, deciduous, and broad-leaved woody plants was significantly correlated with AP, and decreased with increasing AP ($R$ = -0.12, $P$ < 0.001; $R$ = -0.13, $P$ < 0.001; $R$ = -0.11, $P$ < 0.001; $R$ = -0.12, $P$ < 0.001; $R$ = -0.13, $P$ < 0.001) (Fig 3).

### 3.3 Change of nitrogen-phosphorus scaling exponent with environmental gradient

The N-P scaling exponents of woody plants of different life forms differed and were not constrained to a constant value of 2/3 (Table 2). The N-P scaling exponent of all woody plant data collected together was 0.70, and no significant difference with 2/3 in statistics was found. The N-P scaling exponents of trees, shrubs, evergreen, deciduous, and coniferous woody plants were 0.67, 0.72, 0.63, 0.72, and 0.66, respectively. Only the N-P scaling exponent had a significant negative correlation with the annual average temperature ($R$ = -0.20, $P$ < 0.05). No significant correlation was found with annual average precipitation, soil AN, and AP (Fig 4). The scaling exponent of the N-P relationship was found to be positively correlated with LP, and negatively correlated with the N: P ratio, but not significantly correlated with LN (Fig 5).

### 3.4 Relative importance of biological and abiotic factors to leaf stoichiometry

Redundancy analysis (RDA) showed that all predictive variables explained 19.6%, 11.7%, and 8.2% of the changes in LN, LP, and N:P, respectively (Fig 6). Various other sources may also affect leaf stoichiometry variation, such as unquantified microenvironment, species specific variation, soil time series, and interference factors [10]. The relative importance of predictive variables to leaf stoichiometry was ranked in descending order of importance. Among the driving factors affecting LN, the main factor was life form (58.11%), followed by MAP (21.17%) and MAT (19.39%). Among the driving factors affecting LP, the main factor was MAT (41.20%), followed by life form (28.80%) and MAP (14.79%). Among the driving factors

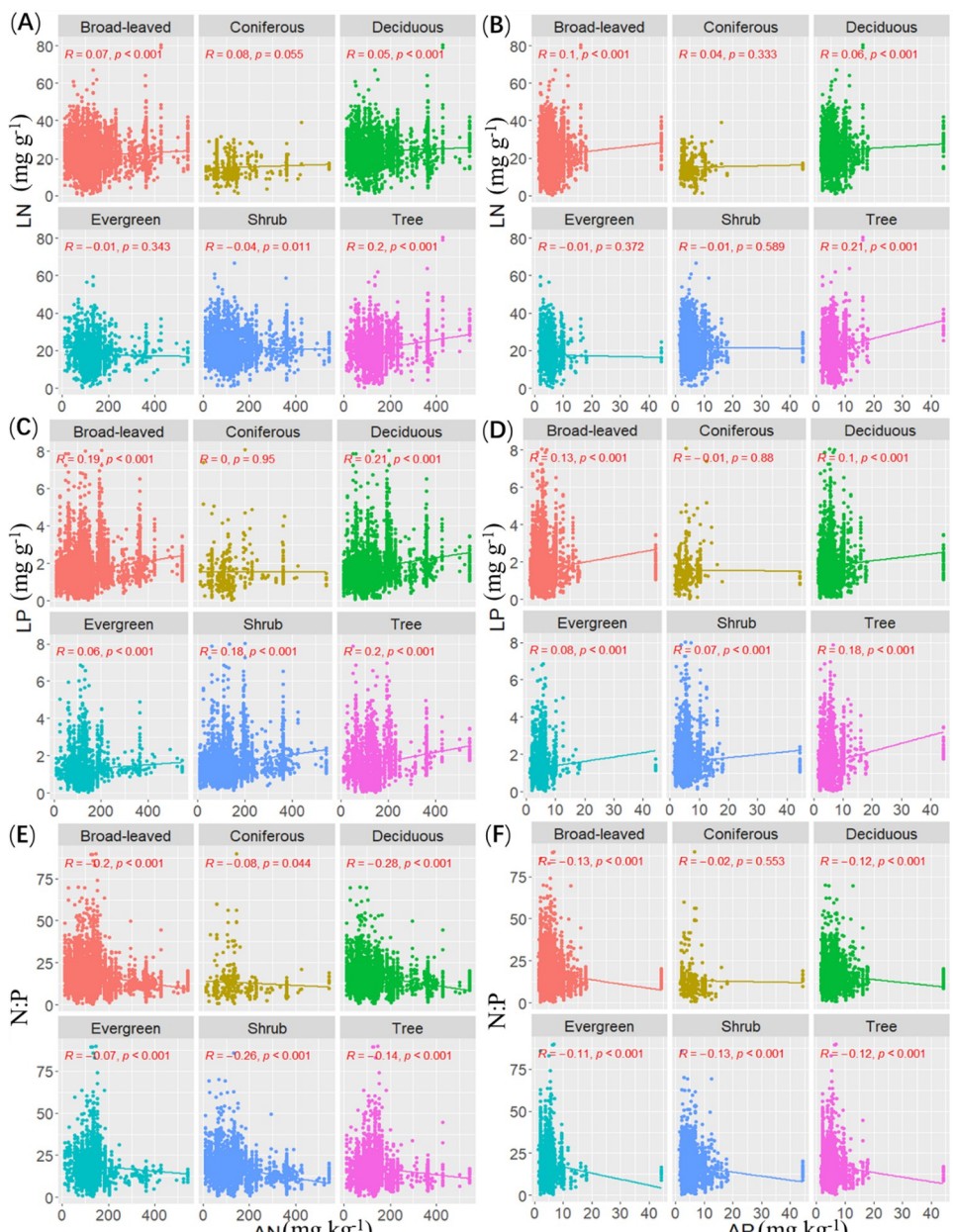

**Fig 3. Relationship between stoichiometry of leaves of different life forms and soil factors.**

affecting N: P, the main factor was AN (36.32%), followed by MAT (34.21%) and life form (12.11%).

## 4 Discussion

In this study, data on N and P content in leaves of Chinese woody plant were utilized. A total of 10,719 data pairs were used to explore whether a general proportion relationship exists between LN and LP. The results indicate that there is no unified 2/3 scaling exponent for the proportional relationship between LN and LP among different functional groups. Therefore, the general 2/3 power law does not apply to different functional groups of Chinese woody plants. This

**Table 2. Summary of the SMA regression results of N and P concentrations in leaves of different functional groups of terrestrial plants.** $\log_{10}LN = \alpha \log_{10}LP + \log_{10}\beta$; "√" indicates that there is no significant difference between N and P scale indexes and 2/3; "<" and ">" indicate that the scaling index of blade N and P is less than or greater than 2/3, respectively. The P value is obtained by likelihood ratio test.

| Form | α | $R^2$ | P | n | 2/3 |
|---|---|---|---|---|---|
| Woody plant | 0.70 | 0.19 | P<0.001 | 10719 | √ |
| Tree | 0.67 | 0.24 | P<0.001 | 5143 | √ |
| Shrub | 0.72 | 0.13 | P<0.001 | 4970 | √ |
| Evergreen woody | 0.63 | 0.09 | P<0.001 | 4344 | < |
| Deciduous woody | 0.72 | 0.17 | P<0.001 | 5772 | > |
| Coniferous woody | 0.66 | 0.17 | P<0.001 | 603 | √ |
| Broad-leaved woody | 0.70 | 0.18 | P<0.001 | 10116 | √ |

value also varies among different sites and is significantly correlated with climatic factors, while soil factors have no significant impact on the scaling exponent (Fig 4A, 4C and 4D).

## 4.1 Stoichiometric characteristics of leaves and changes of N-P scaling exponent

The average LN of Chinese woody plants (20.77 mg g$^{-1}$) found in this study was consistent with the previously reported N content in global terrestrial plants (20.1 mg g$^{-1}$) [22]. However, the average LP (1.58 mg g$^{-1}$) was significantly lower than the previously reported P content in global terrestrial plants (1.80 mg g$^{-1}$) [22]. The scaling exponent (0.70) was consistent with the

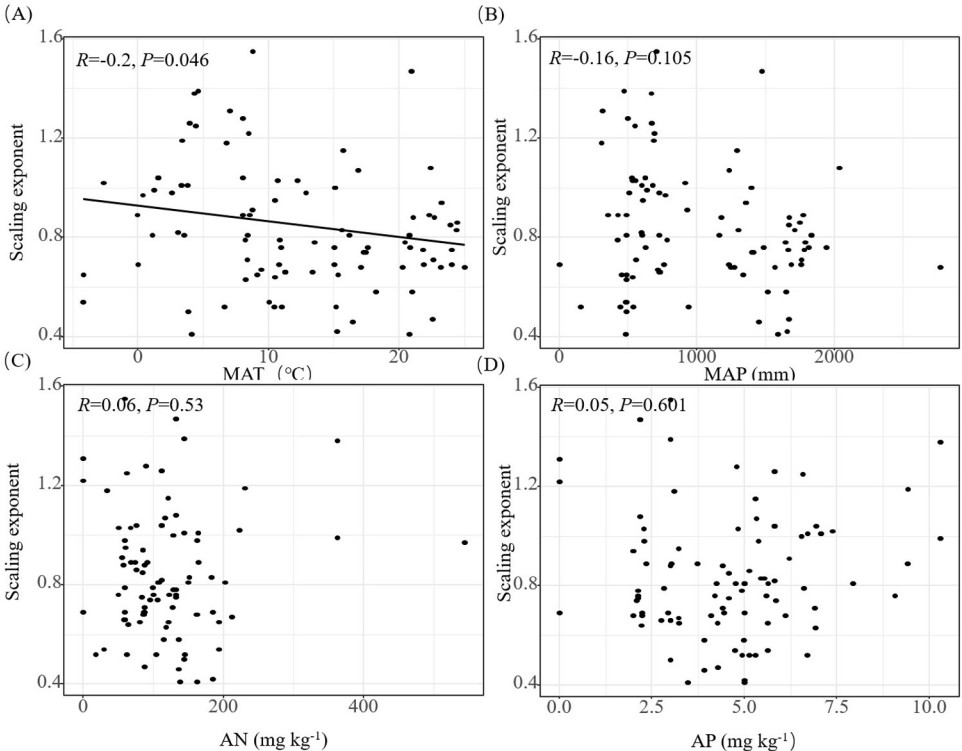

**Fig 4. Relationship between N-P scale index and MAT, MAP, soil AN, and soil AP at different sites.** Climate and soil values are obtained from the geometric average of each station. The proportion index was calculated through SMA regression of leaf nitrogen and phosphorus concentration (α), such as $\log_{10}LN = \alpha \log_{10}LP + \log_{10}\beta$.

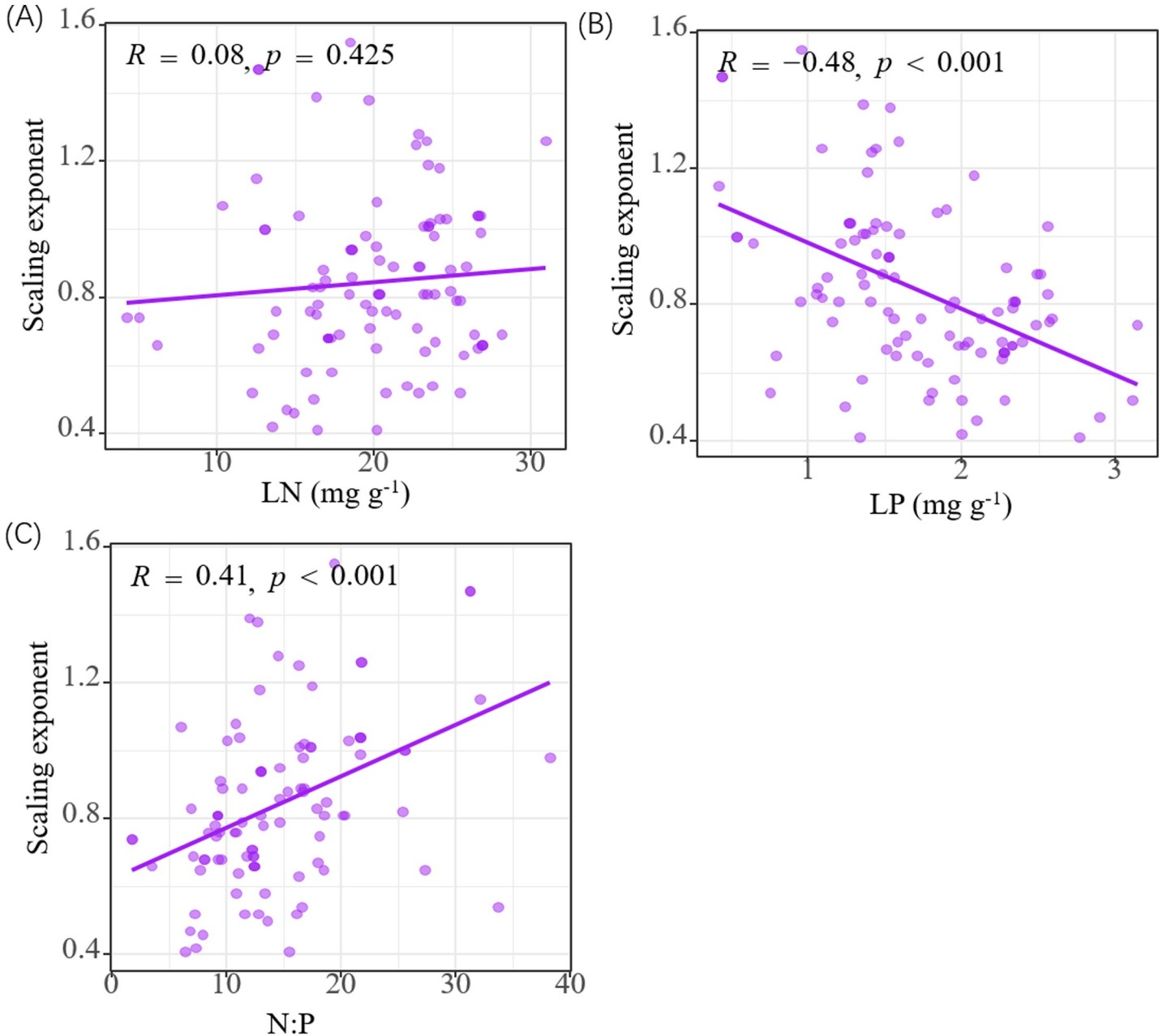

**Fig 5. Relationship between N-P ratio index and N, P, and N: P mass ratio of leaves at different sites.** (a) Relationship between nitrogen and phosphorus ratio index and nitrogen content of leaves; (b) relationship between N: P ratio index and P content in leaves; (c) relationship between N-P proportional index and N: P mass ratio. Leaf N and P contents and N: P ratio are calculated from the geometric mean value of each station. The proportion index was calculated through SMA regression of leaf N and P concentration (α). For example, $\log_{10}LN = \alpha \log_{10}LP + \log_{10} \beta$.

reported global individual level average (0.667) [41], which is close to 2/3 but lower than the Asian level (0.712) [14]. The LN and LP of deciduous plants were found to be significantly higher than those of evergreen plants, which is consistent with previous results [14, 23]. Evergreen trees, distributed at low and middle latitudes, had low soil nutrient availability [56, 57]. Compared with deciduous plants, leaves had longer life spans, higher leaf quality per unit area, and lower photosynthetic rates [58, 59], which may lead to lower nutrient contents in

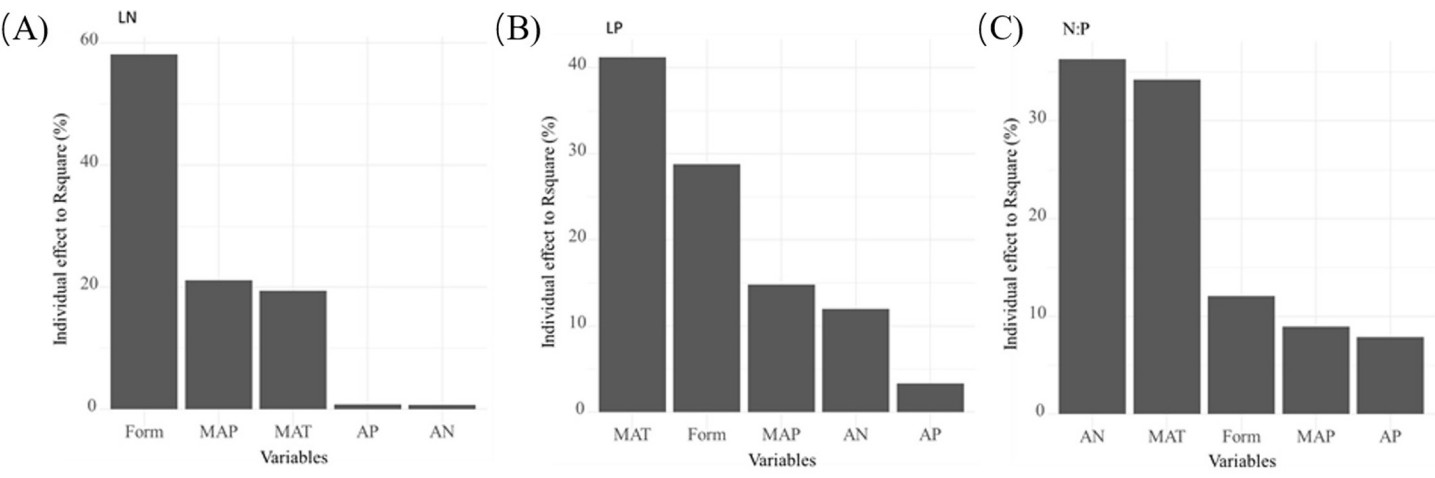

**Fig 6. Relative importance analysis of blade stoichiometry.**

evergreen plant tissues. In contrast, deciduous trees with their shorter leaf life need to maximize leaf photosynthesis in the short growing season, so that more nutrients are distributed to tissues. In this study, the LN of broad-leaved trees was found to be higher than that of conifers [60]. Unlike P, which plants only absorb through their roots, plants absorb N through both their leaves and roots. Broad leaved trees have a higher leaf surface area and can obtain more N absorption pathways, which may lead to higher N contents of broadleaved trees compared to coniferous trees [13]. The N-P scaling exponent was mainly determined by life form and MAT (Table 2 and Fig 4A). The N-P scaling exponent of deciduous plants was also higher than that of evergreen plants (Table 2). Deciduous broad-leaved plants tend to have shorter leaf life-spans and faster leaf growth [61, 62]; therefore, their demand for P is higher, and their N and P levels are lower, while deciduous plants have a lower numerical scaling exponent compared to evergreen broad-leaved plants [62]. These findings are inconsistent with the results obtained in this study, which may indicate that the N-P scaling exponent was also affected by the external environment. For example, more data on evergreen plants were collected in areas with high MAT, which will lead to a low N-P scaling exponent of evergreen plants. In contrast, the N-P scaling exponent values of coniferous forests and broad-leaved woody plants were low, reflecting not only the P-related growth rate, but also the morphological and anatomical characteristics of leaves. The specific leaf area of conifers is generally smaller than that of broad-leaved plants (Table 2) [63], which may lead to a relatively low demand for nutrients, especially N [64].

### 4.2 Effects of climate and soil on N, P, N: P, and N-P scaling exponent of leaves

Compared with LN, LP changed more and was also more closely related to climate and soil (Table 1 and Fig 6A and 6B), which is consistent with previous studies on different plant groups and different geographical regions [25, 65, 66]. LN was mainly determined by the changes of plant growth morphology along the latitude gradient, while leaf P and N: P were determined by both MAT and plant growth morphology (Fig 6A–6C) [10]. In general, the contents of N and P in Chinese woody plants decreased with increasing MAT and MAP; N: P increased with increasing MAT and MAP, which strongly supports the temperature plant physiology hypothesis. These results are similar to previous findings, showing that the concentrations of N and P in

global plant leaves decreases toward the equator [12, 22, 24]. LP of woody plants in China increased with increasing AN and AP, and N: P decreased with increasing AN and AP. There are several reasons for the pattern of P concentrations in plants. First, as the plant metabolic process depends on temperature, under low temperatures, plants usually increase the concentration of P in their cells to compensate for the reduction of their metabolic rate [67]. Second, the P plants absorb mainly originates from the soil, and the content of P in the soil was found to decrease significantly with increasing MAT and MAP in the forest ecosystem. This may also lead to a low content of P in plant tissues in areas with high P content in the forest ecosystem soil. Previous studies on plant leaves showed that wide differences in soil P supply at latitude and biological community level lead to differences in plant P nutrition [65, 68]. Third, soil parent material is the main P source for plant growth. Soils in areas with high MAT and MAP usually have high weathering and leaching characteristics, which may limit P availability for plants in these areas [69]. Therefore, LP in high MAT areas and high MAP areas may be lower than in low MAT areas and low MAP areas. Previous studies have shown that a N: P ratio of leaves below 14 often indicates N limitation, while a N: P ratio exceeding 16 often indicates P limitation [70]. The average N: P ratio found in this study (17.28) may indicate that Chinese woody plants are mainly restricted by P. The leaf N: P (which increased with increasing MAT and MAP) of Chinese forests increased from north to south, indicating that from north to south, Chinese woody plants may transfer from P limitation to N limitation. The results showed that in recent decades, P in forest soil has followed a downward trend in China, which may lead to an increase in the N: P ratio and the restriction of P in forest ecosystems, especially in areas with high N deposition (high MAT and MAP) [71].

Many factors affect the stoichiometric characteristics and spatial distribution pattern of plant leaves, mainly including climatic factors [22, 72], soil characteristics [10, 73], and biological factors (such as plant species and life forms) [34]. The results of this study indicate that when the research subjects involve a large number of plant life forms, these life forms often become a key driving factor for the variation of plant leaf stoichiometric characteristics, and greatly weaken the impact of climatic factors and soil characteristics [34, 35, 66]. In addition, significant differences were found in the stoichiometric characteristics and spatial distribution patterns of leaves among different life forms of plants. For example, the element contents in the leaves of deciduous tree species are significantly higher than that of evergreen tree species [23].

Although the N-P scaling exponent of woody plant leaves in China was approximately 2/3, the N-P scaling exponent varied greatly among different functional plant types and regions. Based on a large data set of global leaf N and P contents (12,055 records), the rule of 2/3 scaling exponent was probed. Large differences were found between different sampling points, latitudes, regions, and functional groups, which may be regulated by plant growth rate, soil N or P availabilities, leaf life-spans, climatic, and other factors [14]. The results of this study showed that the range of N-P scaling exponents of different functional types of woody plants in China was 0.63–0.72, which supports Tian's view that the N-P scaling exponent relationship of leaves is inconstant. The typical value of the global leaf N-P scaling exponent may be the result of analysis of aggregated data, which hide or ignore important biological and ecological changes [14]. Correlation analysis of the N-P scaling exponent with climatic and soil factors showed that it was only significantly negatively correlated with MAT, but not with soil factors (Fig 4A and 4C and 4D). This result is inconsistent with the results of N-P scaling exponent reduction with increasing soil P content at a global scale [14]. Possibly, when the availability of P in the soil is insufficient, plants tend to absorb N excessively (i.e., luxury consumption of N) [74–77], and vice versa [78, 79]. The analysis of leaf N and P content data of 13 forest woody plants in eastern China showed that α fluctuated significantly from 0.57 to 1.42 [10]. Nevertheless, integrating the N and P content data of leaves in TRY showed that the plant N and P contents

follow a relatively constant power index law, as they are globally close to 2/3 across different functional groups [41]. Excessive intake of elements may mask the requirements of chemometrics to varying degrees and greatly change the N-P scaling exponent. From this, it can be inferred that the N-P scaling exponent of leaves was higher because of excessive N absorption by P limited plants; the N-P scaling exponent of leaves was lower because of excessive P absorption by N limited plants [76]. In addition, under the condition of heavy N absorption, LN was occasionally reported to be negatively correlated with LP [80]. However, considering the widespread existence of N and P restriction in terrestrial ecosystems [81, 82], and the difficulty associated with detecting the extent of excessive nutrient absorption, it may be difficult to assess the precise scaling exponent of N and P because of excessive nutrient absorption at different nutrient restricted sampling points [14].

## 5 Conclusion

The results of this study provide detailed leaf N and P concentrations as well as N: P ratios of different functional groups of woody plants in China. The N and P contents of deciduous plants were found to be significantly higher than those of evergreen plants. In general, N and P contents decreased with increasing MAT and MAP, and increased with increasing soil AN and AP. Life form was the main driving factor of N content in leaves, MAT was the main driving factor of P content in leaves, and AN was the main driving factor of the N: P ratio. The N-P scaling exponent of leaves was mainly affected by temperature. The scaling exponent of coniferous trees was the lowest (0.66), while that of deciduous trees was the highest (0.72). In addition, the scaling exponent was found to vary with changing sampling point, and the geometric mean value was 0.85, with a range of 0.41–1.55. These results show that there is no typical value for the scaling exponent of N and P, and the comprehensive data analysis of this scale relationship may conceal significant biological and ecological changes.

## Supporting information

**S1 Data.**
(XLSX)

**S2 Data.**
(XLSX)

## Acknowledgments

We thank other members of the research team for their continuous suggestions and encouragement in the research process.

## Author Contributions

**Data curation:** Xiangguang Duan.

**Formal analysis:** Xiangguang Duan.

**Methodology:** Xiangguang Duan.

**Software:** Xiangguang Duan.

**Visualization:** Xiangguang Duan.

**Writing – original draft:** Xiangguang Duan.

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
