## [Decision Letter · Decision Letter 0]

22 May 2023

PONE-D-23-11129Stoichiometric characteristics of woody plant leaves and responses to climate and soil factors in ChinaPLOS ONE

Dear Dr. Duan,

Thank you for submitting your manuscript to PLOS ONE. After careful consideration, we feel that it has merit but does not fully meet PLOS ONE’s publication criteria as it currently stands. Therefore, we invite you to submit a revised version of the manuscript that addresses the points raised during the review process.

We look forward to receiving your revised manuscript.

Kind regards,

Xiao Guo, Ph.D.

Academic Editor

PLOS ONE

Journal Requirements:

Additional Editor Comments (if provided):

Dear author, based on the comments provided by the two referees, I should inform that your manuscript cannot be accepted for publication in this current form and “major revisions” is needed.

Both reviewers found strengths in your work but they also found several shortcomings that should be carefully addressed before I take my final decision. The reviewers gave excellent suggestions that will certainly improveyour work (see below). Please pay special attention to the major concerns argued by Reviewer # 2, which deal with the data analysis, and argued by Reviewer # 1 and # 2, who complained about the English usage. Reviewer #2 also pointed out the necessity to reorganize parts of the introduction and discussion. I agree with the two reviewers and kindly invite you to provide a revised version of your manuscript considering and responding to the suggestions of all reviewers.  My final decision will be taken after your revised manuscript has been reviewed by reviewers again. 

Reviewers' comments:

Reviewer's Responses to Questions

**Comments to the Author**

1. Is the manuscript technically sound, and do the data support the conclusions?

Reviewer #1: Partly

Reviewer #2: Partly

2. Has the statistical analysis been performed appropriately and rigorously? 

Reviewer #1: Yes

Reviewer #2: No

3. Have the authors made all data underlying the findings in their manuscript fully available?

Reviewer #1: No

Reviewer #2: Yes

4. Is the manuscript presented in an intelligible fashion and written in standard English?

Reviewer #1: Yes

Reviewer #2: No

5. Review Comments to the Author

Reviewer #1: The research topic appears to be of interest and appropriate for publication in PLOS ONE (ISSN 1932-6203). This manuscript examines the content of "Stoichiometric characteristics of woody plant leaves and responses to climate and soil factors in China" (PONE-D-23-11129), and reports the data sets on leaf nitrogen and phosphorus concentrations and N: P ratios in Chinese woody plants have been compiled and classified by life forms.

There are some technical and structural issues in this study that need to be addressed during the revision process.

The following are the details of the comments.

1. Why is there a discrepancy between the author affiliation provided in the journal system and the title page? Regarding product ownership, it is a serious issue.

2. L8: Language quality can be improved here (e.g., ecological stoichiometry) as well as similar minor errors in each section.

3. L23: The standard deviation should be added to these mean values as means + SD.

4. L117-124: There is no clear explanation of the research hypothesis in this study.

5. L117-124: Aims should be categorized into points that correspond to the headings of results and discussion in order to clarify them more clearly.

6. L126-138: The study areas are not adequately described. Although it may be impossible to provide the details of all sites, the author can provide specific information about leading sites.

7. L133: The list along with necessary details of the 413 genera and 939 species used in this study should be provided as supplementary files in order to provide clear information to the reader.

8. L135: The list along with necessary details of the 67 sites used in this study should be provided as supplementary files to provide clear information to the reader.

9. The study does not present any details regarding the research indicators measurements, which is not the appropriate method to present the study.

10. L153-167: The standard deviation should be added to these mean values as means + SD.

11. Table 1: The standard deviation should be added to these mean values as means + SD.

12. Figures 1, 2 and 3: Most of the results are not significant or have very low r values. Is it meaningful to present such weak or insignificant results?

13. Figures 1, 2, 3, and 5: The sub-figures number should be added as it appears in figure 4.

14. 335-337: What is the reason for the inconsistent reference style?

15. 391: There is no information regarding the limitations of the study.

16. 552-577: Is it necessary to include the names of all authors?

17. At certain points in the text, it may be necessary for authors to add appropriate literature in support of the content. The following literature may be considered by authors; lines 39-42: Plant-soil interactions and C:N:P stoichiometric homeostasis of plant organs in riparian plantation (https://doi.org/10.3389/fpls.2022.979023); Effects of hydrological regime on Taxodium ascendens plant decomposition and nutrient dynamics in the Three Gorges Reservoir riparian zone (https://doi.org/10.3389/fenvs.2022.990485) and The effect of hydrological regimes on the concentrations of nonstructural carbohydrates and organic acids in the roots of Salix matsudana in the Three Gorges Reservoir, China (https://doi.org/10.1016/j.ecolind.2022.109176).

Reviewer #2: This study compiled and classified a dataset of leaf nitrogen, phosphorus concentration, and nitrogen phosphorus ratio of Chinese woody plants based on different life forms. The results indicate that the internal attributes of different life forms, the growth rate related to phosphorus and the relative nutrient availability of soil might be the reasons for the unsteady relationship between nitrogen and phosphorus in leaves. Generally, this work is meaningful, but the manuscript by Duan needs to be substantially improved before being accepted for publication in Plos one at this stage. It is not yet sufficiently well written and I identify 3 major obstacles to this:

1/ There are errors in the data analysis method. Table 1 conducted an analysis of variance on tree, shrub, evergreen woody, deciduous woody and coniferous woody, which are not independent and cannot be analyzed in this way. They should be divided into three groups, including ‘trees and shrubs’, ‘deciduous forests and evergreen forest’, ‘coniferous forest and broad-leaved forests’ for independent T-test analysis. Correspondingly, the figures should also be analyzed into these three groups. The information on broad-leaved forests should be supplemented.

2/Some studies have shown that the N and P contents in leaves are greatly influenced by forest age. How did you consider this factor with so much data?

3/ the English language is not sufficiently well mastered so the manuscript is poorly written in English. A correction of the English language by an English-native speaker appears crucial because too many sentences and paragraphs remain incomprehensible even after revision.

Abstract:

Firstly, the abstract has too many words, and the journal requires a maximum of 300 words for the abstract. Then the research background is too long and occupies most of the paragraphs, requiring simplification. And the results need to be further condensed, and the abstract lacks conclusion at the end.

Introduction:

The introduction section needs further modification. The research background lacks the research progress on the stoichiometric characteristics of woody plant leaves in China. Why do you study the nitrogen and phosphorus content of woody plants in China? What is the current research progress and where is the innovation of this research?

Materials and methods:

This section need add a distribution map of sampling points. It is recommended to organize the list of species studied and other information into the supplementary materials section. There are errors in the statistical methods. According to the previous suggestion, the analysis of variance should be changed to a T-test and divided into three groups for analysis. Additionally, information on broad-leaved forests should be supplemented.

Results:

It is important to match the results reported in the table with the analysis. Also, make sure to indicate clearly if the results are significant or not. Tables and figures should be divided into three groups for analysis: ‘trees and shrubs’, ‘deciduous forests and evergreen forest’, ‘coniferous forest and broad-leaved forest’.

Discussion:

Well written but the scope of the discussion does not go far enough. So far, the discussion is merely restating what other studies have already reported. The authors must offer an other aspect to their findings.

Tables and figures:

Table 1: It is recommended to use the form of mean ± SE for the nitrogen and phosphorus content of the data.

Figures: For figures 1 and 2, it is recommended that the format of the P-value be consistent, with some being "p<0.001" and others being "p=0.029". It is recommended to change it to "p<0.05" or "p>0.05". In addition, the P value and R2 should have the same number of decimal places retained. The colors in figures 3 and 4 should be consistent.

6. PLOS authors have the option to publish the peer review history of their article (what does this mean?). If published, this will include your full peer review and any attached files.

Reviewer #1: **Yes: **MUhammad Arif, Southwest University, China

Reviewer #2: No

---

## [Author Response · Author response to Decision Letter 0]

26 Jul 2023

Reply to the editor and reviewers

Manuscript ID: PONE-D-23-11129

Title: Stoichiometric characteristics of woody plant leaves and responses to climate and soil factors in China

Authors: Xiangguang Duan

Dear Ph.D. Guo,

First of all, we should thank you and the reviewers for your expert comments and valuable suggestions to our manuscript. After reading the comments, we came to realize that there were indeed some weak points and unsuitable expression in our original manuscript. The issues raised by you and the two reviewers were of great help for our revision. 

In the past few weeks, we have seriously revised the original manuscript. As you will see, all the questions and suggestions were taken into consideration in the revised manuscript. Below we explain in detail how the questions and suggestions were addressed.

Reply to the reviewer 1

Reviewer #1: Review summary:

The research topic appears to be of interest and appropriate for publication in PLOS ONE (ISSN 1932-6203). This manuscript examines the content of "Stoichiometric characteristics of woody plant leaves and responses to climate and soil factors in China" (PONE-D-23-11129), and reports the data sets on leaf nitrogen and phosphorus concentrations and N: P ratios in Chinese woody plants have been compiled and classified by life forms.

There are some technical and structural issues in this study that need to be addressed during the revision process.

Question 1: Why is there a discrepancy between the author affiliation provided in the journal system and the title page? Regarding product ownership, it is a serious issue

Reply: 

Thank you for your comments. Due to applying to this journal during my doctoral studies at Beijing Forestry University, the information was not updated in a timely manner and has now been revised. My current employer is Xinyang Normal University.

Question 2: L8: Language quality can be improved here (e.g., ecological stoichiometry) as well as similar minor errors in each section.

Reply: 

Thank you for your comments. It has been modified by language polishing company according to your suggestions.

Question 3: L23: The standard deviation should be added to these mean values as means + SD

Reply: 

Thank you for your comments. The standard deviation had been added to these mean values as means + SD (20.77±8.12 mg g-1 and 1.58±1.00 mg g-1).

Question 4: L117-124: There is no clear explanation of the research hypothesis in this study.

Reply: 

Thank you for your comments. It has been modified in the manuscript according to your suggestions. The main purpose of this study is to test the universality of leaf N and P chemometric relationships and the influencing factors of N-P scaling exponent in Chinese woody plant.

Question 5: L117-124: Aims should be categorized into points that correspond to the headings of results and discussion in order to clarify them more clearly.

Reply: Thank you for your comments. It has been modified in the manuscript according to your suggestions. In this paper, the overall chemometrics of leaf N and P are discussed first, as they can be used as a basis for comparison with other studies. Next, the N and P contents and scale indices of different life forms were determined, and it was examined whether numerical changes in stoichiometric characteristics are related to climatic conditions and soil factors. In addition, the numerical change patterns of different site scaling exponents are also discussed to test whether the overall scale relationship between N and P masked significant differences in site correlation.

Question 6: L126-138: The study areas are not adequately described. Although it may be impossible to provide the details of all sites, the author can provide specific information about leading sites.

Reply: 

Thank you for your comments. It has been modified in the manuscript according to your suggestions. The research area covers almost all provinces in China (Fig 1). 

Fig. 1 Sampling locations of leaf nitrogen and phosphorus for all species in this study

Question 7: L133: The list along with necessary details of the 413 genera and 939 species used in this study should be provided as supplementary files in order to provide clear information to the reader.

Reply: 

Thank you for your comments. We have provided a list along with necessary details of the 413 genera and 939 species used in this study as supplementary files according to your suggestions.

Question 8: L135: The list along with necessary details of the 67 sites used in this study should be provided as supplementary files to provide clear information to the reader.

Reply: 

Thank you for your comments. We have provided a list as supplementary files according to your suggestions.

Question 9: The study does not present any details regarding the research indicators measurements, which is not the appropriate method to present the study.

Reply: 

Thank you for your comments. It has been modified in the manuscript according to your suggestions. LN were measured using the methods described by He et al. (2006) [1]. LP was measured after H2SO4-H2O2-HF digestion via the ammonium molybdate/stannous chloride method. Stoichiometric ratios (N/P) were calculated based on dry mass [2].

Question 10: L153-167: The standard deviation should be added to these mean values as means + SD.

Reply: 

Thank you for your comments. It has been modified in the manuscript according to your suggestions.

Question 11: Table 1: The standard deviation should be added to these mean values as means + SD.

Reply: 

Thank you for your comments. It has been modified in the manuscript according to your suggestions.

Question 12: Figures 1, 2 and 3: Most of the results are not significant or have very low r values. Is it meaningful to present such weak or insignificant results?

Reply: 

Thank you for your comments. Due to the inevitable errors in data collection and aggregation, most of the results are not significant and demonstrate that environmental factors have different impacts on different life forms of woody plants. Subsequent research on the family and genus levels also confirms this result. The r value generally decreases with the increase of data volume, so although the r value is small, P<0.05 can indicate its significance.

Question 13: Figures 1, 2, 3, and 5: The sub-figures number should be added as it appears in figure 4.

Reply: 

Thank you for your comments. It has been modified in the manuscript according to your suggestions.

Question 14: 335-337: What is the reason for the inconsistent reference style?

Reply: 

Thank you for your comments. It has been modified in the manuscript according to your suggestions.

Question 15: 391: There is no information regarding the limitations of the study.

Reply:

Thank you for your comments. It has been modified in the manuscript according to your suggestions. Based on the analysis of leaf N and P content data of 13 forest woody plants in eastern China, it was found that α There is a significant fluctuation, ranging from 0.57 to 1.42. Nevertheless, by integrating the N and P content data of leaves in TRY, it was found that α was the plant N and P contents follow a relatively constant power index law, as they are close to 2/3 globally and across different functional groups.

Question 16: 552-577: Is it necessary to include the names of all authors?

Reply: 

Thank you for your comments. Perhaps not necessary, just for formatting purposes. It has been modified in the manuscript according to your suggestions.

Question 17: At certain points in the text, it may be necessary for authors to add appropriate literature in support of the content. The following literature may be considered by authors; lines 39-42: Plant-soil interactions and C:N:P stoichiometric homeostasis of plant organs in riparian plantation (https://doi.org/10.3389/fpls.2022.979023); Effects of hydrological regime on Taxodium ascendens plant decomposition and nutrient dynamics in the Three Gorges Reservoir riparian zone (https://doi.org/10.3389/fenvs.2022.990485) and The effect of hydrological regimes on the concentrations of nonstructural carbohydrates and organic acids in the roots of Salix matsudana in the Three Gorges Reservoir, China (https://doi.org/10.1016/j.ecolind.2022.109176).

Reply: 

Thank you for your comments. It has been modified in the manuscript according to your suggestions. 

Reply to the reviewer 2

Reviewer #2: Review summary:

This study compiled and classified a dataset of leaf nitrogen, phosphorus concentration, and nitrogen phosphorus ratio of Chinese woody plants based on different life forms. The results indicate that the internal attributes of different life forms, the growth rate related to phosphorus and the relative nutrient availability of soil might be the reasons for the unsteady relationship between nitrogen and phosphorus in leaves. Generally, this work is meaningful, but the manuscript by Duan needs to be substantially improved before being accepted for publication in Plos one at this stage. It is not yet sufficiently well written and I identify 3 major obstacles to this: 

Question 1: There are errors in the data analysis method. Table 1 conducted an analysis of variance on tree, shrub, evergreen woody, deciduous woody and coniferous woody, which are not independent and cannot be analyzed in this way. They should be divided into three groups, including ‘trees and shrubs’, ‘deciduous forests and evergreen forest’, ‘coniferous forest and broad-leaved forests’ for independent T-test analysis. Correspondingly, the figures should also be analyzed into these three groups. The information on broad-leaved forests should be supplemented.

Reply: 

Thank you for your comments. It has been modified in the manuscript according to your suggestions.

Form n LN mean(mg g−1) LP mean(mg g−1) N:P 　

Woody plant 

10719 20.77 (8.12) 1.58 (1.00) 17.28 (22.40)

Tree 5143 20.56 (8.12) b 1.56 (1.00) a 17.22 (9.59) a

Shrub 4970 21.57 (8.06) a 1.60 (1.00) a 16.92 (8.31) a

Evergreen woody 4344 17.43 (6.77) b 1.28 (0.92) b 18.33 (10.32) a

Deciduous woody 5772 23.81 (7.95) a 1.80 (1.00) a 16.11 (7.69) b

Coniferous woody 603 14.98 (5.66) b 1.54 (0.96) a 12.94 (9.45) b

broad-leaved

woody 10116 20.78 (8.12) a 1.58 (1.00) a 17.28 (22.41) a

Question 2: Some studies have shown that the N and P contents in leaves are greatly influenced by forest age. How did you consider this factor with so much data?

Reply: 

Thank you for your comments. Some studies have shown that the N and P contents in leaves are greatly influenced by forest age[3]. However, in most of these studies across a large scale, age sequence was not considered. Photosynthetic capacity[4] and nutrient requirements[5] generally vary with plant growth, especially for woody plants or trees. Trees at different growth stages have huge differences in physiological processes and nutrition requirements, resulting in tree nutrient stoichiometry changes along an age sequence[5, 6]. However, it remains unclear whether age variable is a determinant factor to the patterns of tree stoichiometry across a large scale. Moreover, the research data on the impact of forest age on leaf chemometrics is insufficient, leading to difficulties in big data analysis. 

Question 3: the English language is not sufficiently well mastered so the manuscript is poorly written in English. A correction of the English language by an English-native speaker appears crucial because too many sentences and paragraphs remain incomprehensible even after revision.

Reply: 

Thank you for your comments. It has been modified in the manuscript according to your suggestions. 

Question 4: Introduction: The introduction section needs further modification. The research background lacks the research progress on the stoichiometric characteristics of woody plant leaves in China. Why do you study the nitrogen and phosphorus content of woody plants in China? What is the current research progress and where is the innovation of this research?

Reply: 

Thank you for your comments. It has been modified in the introduction according to your suggestions. Despite extensive research on the relationship between traits in the plant kingdom, our understanding of the coordination of traits at the level of woody organisms lags behind that of herbaceous species [7]. Previous studies have identified a set of key functional traits that summarize the form and functional spectrum of the entire plant kingdom, with leaf economy and plant size being the main trait axes supporting life cycle strategies [7]. Due to the size, life span, individual development and unique structural characteristics of woody plants, compared with herbaceous plants, trees have obvious characteristics and face new abiotic stress [8]. Our current understanding of dominant trait patterns in plants fundamentally overlooks the significant energy investment structure unique to large woody species [8]. Many studies have validated some assumptions about the relationship between nitrogen, phosphorus, and environmental factors at regional and global scales [9]. However, most studies in China have not examined the different responses of leaf N, P, and N: P to environmental variables and plant growth forms [9]. This article explores the response of leaf N, P, and N: P to potential driving factors by collecting leaf nitrogen and phosphorus content data from 413 genera and 939 species of woody plants in China.

Question 5: Materials and methods: This section need add a distribution map of sampling points. It is recommended to organize the list of species studied and other information into the supplementary materials section. There are errors in the statistical methods. According to the previous suggestion, the analysis of variance should be changed to a T-test and divided into three groups for analysis. Additionally, information on broad-leaved forests should be supplemented.

Reply: 

Thank you for your comments. It has been modified in the manuscript according to your suggestions.

Question 6: Results: It is important to match the results reported in the table with the analysis. Also, make sure to indicate clearly if the results are significant or not. Tables and figures should be divided into three groups for analysis: ‘trees and shrubs’, ‘deciduous forests and evergreen forest’, ‘coniferous forest and broad-leaved forest’. 

Reply: 

Thank you for your comments. It has been modified in the manuscript according to your suggestions. 

Question 7: Discussion: Well written but the scope of the discussion does not go far enough. So far, the discussion is merely restating what other studies have already reported. The authors must offer an other aspect to their findings.

Reply: 

Thank you for your comments. It has been modified in the manuscript according to your suggestions.

There are many factors that affect the stoichiometric characteristics and spatial distribution pattern of plant leaves, mainly including climate factors[10, 11], soil characteristics[9, 12], and biological factors (such as plant species, life forms, etc.)[13]. This study indicated that when the research subjects involve a large number of plant life forms, life forms often become a key driving factor for the variation of plant leaf stoichiometric characteristics, and greatly weaken the impact of climate factors and soil characteristics[1, 13, 14]. In addition, there are significant differences in the stoichiometric characteristics and spatial distribution patterns of leaves among different life forms of plants. For example, the element content in the leaves of deciduous tree species is significantly higher than that of evergreen tree species[15].

Question 8: Tables and figures:Table 1: It is recommended to use the form of mean ± SE for the nitrogen and phosphorus content of the data.

Figures: For figures 1 and 2, it is recommended that the format of the P-value be consistent, with some being "p<0.001" and others being "p=0.029". It is recommended to change it to "p<0.05" or "p>0.05". In addition, the P value and R2 should have the same number of decimal places retained. The colors in figures 3 and 4 should be consistent.

Reply:

Thank you for your comments. It has been modified in the manuscript according to your suggestions.

Overall, you will see that we have seriously revised the manuscript based on the new results. We are confident that our manuscript has been much improved. We sincerely hope that this revised manuscript can be finally accepted for publication.

Reference

1. He, J.S.; Fang, J.; Wang, Z.; Guo, D.; Flynn, D.F.; Geng, Z. Stoichiometry and large-scale patterns of leaf carbon and nitrogen in the grassland biomes of China. Oecologia. 2006. 149 (1), 115-22.

2. Sun, L.; Zhang, B.; Wang, B.; Zhang, G.; Zhang, W.; Zhang, B.; Chang, S.; Chen, T.; Liu, G. Leaf elemental stoichiometry of Tamarix Lour. species in relation to geographic, climatic, soil, and genetic components in China. Ecological Engineering. 2017. 106, 448-457.

3. Zhang, H.; Sun, M.; Wen, Y.; Tong, R.; Wang, G.; Wu, Q.; Li, Y.; Wu, T. The effects of stand age on leaf N:P cannot be neglected: A global synthesis. Forest Ecology and Management. 2022. 518.

4. Zheng, Y.; Zhao, Z.; Zhou, J.-J.; Zhou, H.; Liang, Z.-S.; Luo, Z.-B. The importance of slope aspect and stand age on the photosynthetic carbon fixation capacity of forest: a case study with black locust (Robinia pseudoacacia) plantations on the Loess Plateau. Acta Physiologiae Plantarum. 2010. 33 (2), 419-429.

5. Zhang, H.; Guo, W.; Yu, M.; Wang, G.G.; Wu, T. Latitudinal patterns of leaf N, P stoichiometry and nutrient resorption of Metasequoia glyptostroboides along the eastern coastline of China. Sci Total Environ. 2018. 618, 1-6.

6. Yan, T.; Lü, X.-T.; Zhu, J.-J.; Yang, K.; Yu, L.-Z.; Gao, T. Changes in nitrogen and phosphorus cycling suggest a transition to phosphorus limitation with the stand development of larch plantations. Plant and Soil. 2017. 422 (1-2), 385-396.

7. Maynard, D.S.; Bialic-Murphy, L.; Zohner, C.M.; Averill, C.; van den Hoogen, J.; Ma, H.; Mo, L.; Smith, G.R.; Acosta, A.T.R.; Aubin, I.; Berenguer, E.; Boonman, C.C.F.; Catford, J.A.; Cerabolini, B.E.L.; Dias, A.S.; Gonzalez-Melo, A.; Hietz, P.; Lusk, C.H.; Mori, A.S.; Niinemets, U.; Pillar, V.D.; Pinho, B.X.; Rosell, J.A.; Schurr, F.M.; Sheremetev, S.N.; da Silva, A.C.; Sosinski, E.; van Bodegom, P.M.; Weiher, E.; Bonisch, G.; Kattge, J.; Crowther, T.W. Global relationships in tree functional traits. Nat Commun. 2022. 13 (1), 3185.

8. Li, J.; Chen, X.; Niklas, K.J.; Sun, J.; Wang, Z.; Zhong, Q.; Hu, D.; Cheng, D. A whole‐plant economics spectrum including bark functional traits for 59 subtropical woody plant species. Journal of Ecology. 2021. 110 (1), 248-261.

9. Chen, Y.; Han, W.; Tang, L.; Tang, Z.; Fang, J. Leaf nitrogen and phosphorus concentrations of woody plants differ in responses to climate, soil and plant growth form. Ecography. 2013. 36, 178-184.

10. Reich, P.B.; Oleksyn, J. Global patterns of plant leaf N and P in relation to temperature and latitude. Proceedings of the National Academy of Sciences of the United States of America. 2004. 101, 11001-11006.

11. Yuan, Z.Y.; Chen, H.Y.H. Global-scale patterns of nutrient resorption associated with latitude, temperature and precipitation. Global Ecology and Biogeography. 2009. 18 (1), 11-18.

12. Zhang, Y.; Li, C.; Wang, M. Linkages of C: N: P stoichiometry between soil and leaf and their response to climatic factors along altitudinal gradients. Journal of Soils and Sediments. 2018. 19 (4), 1820-1829.

13. Zhao, W.; Reich, P.B.; Yu, Q.; Zhao, N.; Yin, C.; Zhao, C.; Li, D.; Hu, J.; Li, T.; Yin, H.; Liu, Q. Shrub type dominates the vertical distribution of leaf C:N:P stoichiometry across an extensive altitudinal gradient. Biogeosciences. 2018. 15 (7), 2033-2053.

14. He, J.S.; Wang, L.; Flynn, D.F.; Wang, X.; Ma, W.; Fang, J. Leaf nitrogen:phosphorus stoichiometry across Chinese grassland biomes. Oecologia. 2008. 155 (2), 301-10.

15. Han, W.; Fang, J.; Guo, D.; Zhang, Y. Leaf nitrogen and phosphorus stoichiometry across 753 terrestrial plant species in China. New Phytologist. 2005. 168 (2), 377-385.

---

## [Decision Letter · Decision Letter 1]

16 Aug 2023

PONE-D-23-11129R1Stoichiometric characteristics of woody plant leaves and responses to climate and soil factors in ChinaPLOS ONE

Dear Dr. Duan,

Thank you for submitting your manuscript to PLOS ONE. After careful consideration, we feel that it has merit but does not fully meet PLOS ONE’s publication criteria as it currently stands. Therefore, we invite you to submit a revised version of the manuscript that addresses the points raised during the review process.

We look forward to receiving your revised manuscript.

Kind regards,

Xiao Guo, Ph.D.

Academic Editor

PLOS ONE

Additional Editor Comments (if provided):

Although all the previous comments have been thoroughly addressed, Reviewer #2 has made additional valuable suggestions that can further enhance the manuscript. I concur with Reviewer #2's comments. Please revise the manuscript in accordance with these valuable comments. I am looking forward to reviewing the revised manuscript.

Reviewers' comments:

Reviewer's Responses to Questions

**Comments to the Author**

1. If the authors have adequately addressed your comments raised in a previous round of review and you feel that this manuscript is now acceptable for publication, you may indicate that here to bypass the “Comments to the Author” section, enter your conflict of interest statement in the “Confidential to Editor” section, and submit your "Accept" recommendation.

Reviewer #1: All comments have been addressed

Reviewer #2: All comments have been addressed

2. Is the manuscript technically sound, and do the data support the conclusions?

Reviewer #1: Yes

Reviewer #2: Partly

3. Has the statistical analysis been performed appropriately and rigorously? 

Reviewer #1: Yes

Reviewer #2: Yes

4. Have the authors made all data underlying the findings in their manuscript fully available?

Reviewer #1: Yes

Reviewer #2: Yes

5. Is the manuscript presented in an intelligible fashion and written in standard English?

Reviewer #1: Yes

Reviewer #2: Yes

6. Review Comments to the Author

Reviewer #1: While revising, the author followed almost all of the comments and made significant progress.

Reviewer #2: There are still many issues with the manuscript that need further revision, especially with regards to the figures and tables.

1. Figure 1: The map scale is out of balance and there are too many blank spaces. Please adjust it.

2. L194: Table1 “Different letters (such as a, b, and c) represent significant differences (p < 0.05)” There is no ‘c’ in the significance difference.

3. Table 1: The T-test did not indicate in the table which two groups were between, and the representation was incorrect, so it should not be represented in a single row.

4. Table 1: I don't understand what the numbers in parentheses represent, they should be expressed as mean±SE.

5. Figure 3: Why is the comparison of broad-leaved tree species not included in the figure?

6. All fonts in the figures should be consistent, such as inconsistent fonts in Figures 4 and 5.

7. The serial numbers of each small image in Figure 5 are in lowercase letters, while in other images, uppercase letters are used. Please unify them.

8. Discussion: L316: Please indicate which one is in Figure 4A, B, C or D? The same problem also exists in L337, 357…….

7. PLOS authors have the option to publish the peer review history of their article (what does this mean?). If published, this will include your full peer review and any attached files.

Reviewer #1: **Yes: **Muhammad Arif

Reviewer #2: No

---

## [Author Response · Author response to Decision Letter 1]

24 Aug 2023

Reviewer #1: Review summary:

While revising, the author followed almost all of the comments and made significant progress.

Reviewer #2: Review summary:

There are still many issues with the manuscript that need further revision, especially with regards to the figures and tables.

Question 1: Figure 1: The map scale is out of balance and there are too many blank spaces. Please adjust it.

Reply: 

Thank you for your comments. It has been modified according to your suggestions.

Fig. 1 Chinese sampling locations of leaf nitrogen and phosphorus levels for all species in this study

Question 2: L194: Table1 “Different letters (such as a, b, and c) represent significant differences (p < 0.05)” There is no ‘c’ in the significance difference

Reply: 

Thank you for your comments. It has been modified according to your suggestions. 

Table 1. Summary of the statistics of N and P concentrations and N: P ratios in leaves of different functional groups of terrestrial plants. Multiple comparisons were made for each group of trees and shrubs, evergreen and deciduous, coniferous and broad-leaved. Mean represents the geometric mean, and n represents the number of observations. Different letters (such as a and b) represent significant differences (p < 0.05).

Question 3: Table 1: The T-test did not indicate in the table which two groups were between, and the representation was incorrect, so it should not be represented in a single row

Reply: 

Thank you for your comments. It has been modified according to your suggestions.

Table 1. Summary of the statistics of N and P concentrations and N: P ratios in leaves of different functional groups of terrestrial plants. Multiple comparisons were made for each group of trees and shrubs, evergreen and deciduous, coniferous and broad-leaved. Mean represents the geometric mean, and n represents the number of observations. Different letters (such as a and b) represent significant differences (p < 0.05).

Form n LN mean(mg g−1) LP mean(mg g−1) N:P 　

Woody plant 

10719 20.77 ±8.12 1.58 ±1.00 17.28 ±22.40 

Tree 5143 20.56 ±8.12 b 1.56 ±1.00 a 17.22 ±9.59 a

Shrub 4970 21.57 ±8.06 a 1.60 ±1.00 a 16.92 ±8.31 a

Evergreen woody 4344 17.43 ±6.77 b 1.28 ±0.92 b 18.33 ±10.32 a

Deciduous woody 5772 23.81 ±7.95 a 1.80 ±1.00 a 16.11 ±7.69 b

Coniferous woody 603 14.98 ±5.66 b 1.54 ±0.96 a 12.94 ±9.45 b

Broad-leaved

woody 10116 20.78 ±8.12 a 1.58 ±1.00 a 17.28 ±22.41 a

Question 4: Table 1: I don't understand what the numbers in parentheses represent, they should be expressed as mean±SE.

Reply: 

Thank you for your comments. It has been modified in the manuscript according to your suggestions.

Table 1. Summary of the statistics of N and P concentrations and N: P ratios in leaves of different functional groups of terrestrial plants. Multiple comparisons were made for each group of trees and shrubs, evergreen and deciduous, coniferous and broad-leaved. Mean represents the geometric mean, and n represents the number of observations. Different letters (such as a and b) represent significant differences (p < 0.05).

Form n LN mean(mg g−1) LP mean(mg g−1) N:P 　

Woody plant 

10719 20.77 ±8.12 1.58 ±1.00 17.28 ±22.40 

Tree 5143 20.56 ±8.12 b 1.56 ±1.00 a 17.22 ±9.59 a

Shrub 4970 21.57 ±8.06 a 1.60 ±1.00 a 16.92 ±8.31 a

Evergreen woody 4344 17.43 ±6.77 b 1.28 ±0.92 b 18.33 ±10.32 a

Deciduous woody 5772 23.81 ±7.95 a 1.80 ±1.00 a 16.11 ±7.69 b

Coniferous woody 603 14.98 ±5.66 b 1.54 ±0.96 a 12.94 ±9.45 b

Broad-leaved

woody 10116 20.78 ±8.12 a 1.58 ±1.00 a 17.28 ±22.41 a

Question 5: Figure 3: Why is the comparison of broad-leaved tree species not included in the figure?

Reply: Thank you for your comments. It has been modified in the manuscript according to your suggestions.

Fig. 2 Relationship between stoichiometry of leaves of different life forms and climatic factors

Fig. 3 Relationship between stoichiometry of leaves of different life forms and soil factors

Question 6: All fonts in the figures should be consistent, such as inconsistent fonts in Figures 4 and 5.

Reply: 

Thank you for your comments. It has been modified in the manuscript according to your suggestions. 

Fig. 4 Relationship between N-P scale index and MAT, MAP, soil AN, and soil AP at different sites. Climate and soil values are obtained from the geometric average of each station. The proportion index was calculated through SMA regression of leaf nitrogen and phosphorus concentration (α), such as log10LN = α log10LP + log10 β.

Fig. 5 Relationship between N-P ratio index and N, P, and N: P mass ratio of leaves at different sites. (a) Relationship between nitrogen and phosphorus ratio index and nitrogen content of leaves; (b) relationship between N: P ratio index and P content in leaves; (c) relationship between N-P proportional index and N: P mass ratio. Leaf N and P contents and N: P ratio are calculated from the geometric mean value of each station. The proportion index was calculated through SMA regression of leaf N and P concentration (α). For example, log10LN = α log10LP + log10 β.

Question 7: The serial numbers of each small image in Figure 5 are in lowercase letters, while in other images, uppercase letters are used. Please unify them

Reply: 

Thank you for your comments. It has been modified in the manuscript according to your suggestions.



Fig. 5 Relationship between N-P ratio index and N, P, and N: P mass ratio of leaves at different sites. (a) Relationship between nitrogen and phosphorus ratio index and nitrogen content of leaves; (b) relationship between N: P ratio index and P content in leaves; (c) relationship between N-P proportional index and N: P mass ratio. Leaf N and P contents and N: P ratio are calculated from the geometric mean value of each station. The proportion index was calculated through SMA regression of leaf N and P concentration (α). For example, log10LN = α log10LP + log10 β.

Question 8: Discussion: L316: Please indicate which one is in Figure 4A, B, C or D? The same problem also exists in L337, 357…….

Reply: 

Thank you for your comments. It has been modified in the manuscript according to your suggestions.

This value also varies among different sites and is significantly correlated with climatic factors, while soil factors have no significant impact on the scaling exponent (Fig. 4A, 4C and 4D).

The N-P scaling exponent was mainly determined by life form and MAT (Table 2, Fig. 4A).

Compared with LN, LP changed more and was also more closely related to climate and soil (Table 1, Fig. 6A and 6B)

LN was mainly determined by the changes of plant growth morphology along the latitude gradient, while leaf P and N: P were determined by both MAT and plant growth morphology (Fig. 6A, 6B and 6C).

Correlation analysis of the N-P scaling exponent with climatic and soil factors showed that it was only significantly negatively correlated with MAT, but not with soil factors (Fig. 4A, 4Cand 4D).



Overall, you will see that I have seriously revised the manuscript based on the new results. I am confident that my manuscript has been much improved. I sincerely hope that this revised manuscript can be finally accepted for publication.

---

## [Editor Report · Decision Letter 2]

10 Sep 2023

Stoichiometric characteristics of woody plant leaves and responses to climate and soil factors in China

PONE-D-23-11129R2

Dear Dr. Duan,

We’re pleased to inform you that your manuscript has been judged scientifically suitable for publication and will be formally accepted for publication once it meets all outstanding technical requirements.

Kind regards,

Xiao Guo, Ph.D.

Academic Editor

PLOS ONE

Additional Editor Comments (optional):

The authors have diligently revised the manuscript, significantly improving its quality. In the previous round of revisions, Reviewer 2 once again provided excellent suggestions, primarily concerning the annotation and formatting issues of the images and tables. We acknowledge the importance of these detailed matters and sincerely appreciate the thoroughness of Reviewer 2's comments. In this latest submission, the authors have effectively addressed these concerns. Additionally, Reviewer 1 also provided valuable comments, which have been appropriately addressed and approved by Reviewer 1 in the previous round of revision. Based on this, I recommend accepting the manuscript for publication.
---

## [Editor Report · Acceptance letter]

13 Sep 2023

PONE-D-23-11129R2 

Stoichiometric characteristics of woody plant leaves and responses to climate and soil factors in China 

Dear Dr. Duan:

I'm pleased to inform you that your manuscript has been deemed suitable for publication in PLOS ONE. Congratulations! Your manuscript is now with our production department. 

Kind regards, 

on behalf of

Dr. Xiao Guo 

Academic Editor

PLOS ONE